# Clarithromycin Versus Metronidazole in First-Line *Helicobacter Pylori* Triple Eradication Therapy Based on Resistance to Antimicrobial Agents: Meta-Analysis

**DOI:** 10.3390/jcm9020543

**Published:** 2020-02-17

**Authors:** Masaki Murata, Mitsushige Sugimoto, Hitomi Mizuno, Takeshi Kanno, Kiichi Satoh

**Affiliations:** 1Department of Gastroenterology, Shiga University of Medical Science Hospital, Otsu, Shiga 520-2192, Japan; mura0531@belle.shiga-med.ac.jp; 2Department of Gastroenterology, National Hospital Organization Kyoto Medical Center, Fushimi, Kyoto 612-8555, Japan; 3Division of Digestive Endoscopy, Shiga University of Medical Science Hospital, Otsu, Shiga 520-2192, Japan; 4Department of Gastroenterological Endoscopy, Tokyo Medical University Hospital, Shinjuku, Tokyo 160-0023, Japan; 5Toyoda Aoba Clinic, Iwata, Shizuoka 438-0821, Japan; hito0215@toyodaaobaclinic.com; 6Division of Gastroenterology, Tohoku University Hospital, Sendai, Miyagi 980-8578, Japan; kanno.takeshi@med.tohoku.ac.jp; 7Department of Gastroenterology, International University of Health and Welfare Hospital, Nasushiobara, Tochigi 329-2763, Japan; kiichi@iuhw.ac.jp

**Keywords:** *Helicobacter pylori*, eradication therapy, clarithromycin, metronidazole, triple therapy, resistant

## Abstract

Background: International treatment guidelines for *Helicobacter pylori* infection recommend a proton pump inhibitor (PPI)/amoxicillin/clarithromycin (CAM) regimen (PAC) or PPI/amoxicillin/metronidazole (MNZ) regimen (PAM) as first-line therapy based on culture and sensitivity testing. As incidence rates of antimicrobial agent-resistant strains are changing year by year, it is important to reevaluate the efficacy of eradication regimens. We performed a meta-analysis to evaluate the efficacy and safety of PAC and PAM based on different locations categorized by the reported incidence of CAM- and MNZ-resistant strains. Methods: Randomized control trials (RCTs) comparing eradication rates between PAC and PAM first-line treatment up to December 2018 were included. We divided RCTs into four groups based on resistance to CAM (< 15% or ≥ 15%) and MNZ (< 15% or ≥ 15%). Results: A total of 27 studies (4825 patients) were included. Overall eradication rates between PAC and PAM were similar (74.8% and 72.5%, relative risk (RR): 1.13, 95% confidence interval (CI): 0.91–1.39, *P* = 0.27) in the intention-to-treat analysis. In areas with low MNZ- and high CAM-resistance rates, PAM had a significantly higher eradication rate than PAC (92.5% vs. 70.8%, RR: 0.29, 95% CI: 0.13–0.68). In areas with high MNZ- and low CAM-resistance rates, the eradication rate with PAC was only 72.9%. Conclusions: Overall eradication rates with PAC and PAM were equivalent worldwide. In low MNZ-resistance areas, PAM may be recommended as first-line therapy. However, the efficacy of PAC may be insufficient, irrespective of susceptibility to CAM.

## 1. Introduction

The Maastricht V/Florence Consensus Report issued by the European *Helicobacter* Study Group in 2017 recommends eradication of *Helicobacter pylori* (*H. pylori*) for patients with peptic ulcer disease, autoimmune thrombocytopenia, chronic urticaria, iron deficiency anemia, gastric mucosa associated-lymphoid tissue lymphoma, atrophic gastritis, intestinal metaplasia, and functional dyspepsia [1,2,3,4,5,6]. In general, although first-line *H. pylori* eradication triple therapy consisting of a proton pump inhibitor (PPI) and two kinds of antimicrobial agents (e.g., clarithromycin (CAM), amoxicillin (AMPC), metronidazole (MNZ), or levofloxacin) is practiced worldwide, appropriate eradication with high efficacy and safety remains somewhat elusive.

In patients living in areas with high CAM resistance rates (> 15%), such as Japan, the Maastricht V/Florence Consensus Report suggests that when bacterial culture and sensitivity testing are not performed before eradication therapy, first-line triple eradication therapy with PPI and CAM should not be used; rather, bismuth quadruple (PPI, bismuth, tetracycline, and MNZ) or non-bismuth quadruple, concomitant (PPI, AMPC, CAM and nitroimidazole/MNZ) therapies are recommended as the first-line therapy [1].

From 1990 to 2000, eradication rates achieved in Japan using CAM-containing triple therapy ranged from approximately 85%–91% [7]. This rate subsequently decreased with the emergence of CAM-resistant strains [8,9,10,11], and recent patients with these strains have experienced marked decreases, to 10%–30% [12,13]. The most recently measured frequencies of CAM-resistant strains in Japan and Europe exceed 35% and 20%, respectively [7,13,14,15]. This in turn necessitates the careful selection of eradication regimens based on individual antibiotic resistance to *H. pylori* and/or known regional characteristics [1,16]. It is recommended that CAM sensitivity testing be performed when a standard CAM-containing triple regimen (i.e., PPI, AMPC and CAM regimen (PAC regimen)) is considered for first-line therapy, except in populations or regions with well-documented low CAM resistance (<15%) [1]. In addition, in geographical areas where MNZ resistance is almost negligible, replacing CAM with MNZ in triple therapy (i.e., PPI, AMPC and MNZ regimen (PAM regimen)) shows excellent cure rates [17,18]. The Maastricht V/Florence Consensus Report currently recommends the PAM regimen in settings with high CAM resistance. 

Since 2013, *H. pylori* eradication therapy has been approved for all patients in Japan with *H. pylori*-related gastritis confirmed by endoscopy [19]. However, the official first-line eradication regimen is the PAC regimen and the second-line regimen is PAM, irrespective of whether the patient is infected with a CAM- or MNZ-resistant strain of *H. pylori*. Many randomized control trials (RCTs) have compared the efficacy of PAC and PAM, but the year-by-year changes in the incidence rates of antimicrobial resistance mandate that the efficacy of these regimens be periodically reevaluated. In addition, although international guidelines suggest that the selection of treatment regimen be performed based on susceptibility to the antimicrobial agents involved [1], no meta-analysis has yet investigated the association of the efficacy of PAC vs. PAM therapy and geographical area based on incidence rates of antimicrobial agent resistance. 

Here, we performed a meta-analysis to reevaluate the efficacy and safety of PAC and PAM therapies in relation to CAM/MNZ resistance-defined location.

## 2. Materials and Methods

### 2.1. Search Strategy and Inclusion Criteria

In this meta-analysis, we compared eradication rates between CAM-containing PAC therapy and MNZ-containing PAM therapy (each given over 7–14 days) as first-line therapy by conducting a search of the medical literature using data of randomized control trials.

Three researchers (MM, MS and HM) independently searched both the PubMed and Cochrane Library databases using the terms “*Helicobacter pylori*,” “eradication,” and “triple” and reviewed titles and abstracts for all potential studies (Figure 1). 

The inclusion criteria were:RCTs published up to December 2018;Studies comparing eradication rates of PAC therapy with those of PAM therapy for *H. pylori* infection;Studies performed as first-line treatment;Studies that detected *H. pylori* infection by one or more tests (urea breath test, histology, rapid urease test, stool monoclonal antigen or culture);Studies checking eradication outcome ≥ 4 weeks after eradication therapy; andStudies written in English.

Exclusion criteria were:Studies performed with regimens other than PAC or PAM regimens, andStudies with treatment periods > 14 days.

Author names, publication year, country where the study was conducted, number of patients, eradication rate for each regimen, patient characteristics (sex and age), and incidence of adverse events (e.g., diarrhea, skin rash, dysgeusia, and nausea) were extracted from each study.

### 2.2. Statistical Analysis

First, a meta-analysis of RCTs comparing the cure rates and adverse events of PAC versus PAM therapy was performed. For each comparison, intention-to-treat (ITT) and per-protocol (PP) analysis of eradication rates were calculated. Relative risks (RR) and their corresponding 95% confidence intervals (CIs) were used to summarize the effect of each comparison tested using random-effect models, and the calculated results were confirmed in a fixed-effects model as well [20,21,22]. We also divided studies into four groups based on CAM (< 15% or ≥ 15%) and MNZ resistance rates (< 15% or ≥ 15%) in the country where the study was conducted (Appendix A) [10,11,23,24,25,26]. 

Potential study bias in each study was evaluated by funnel plots. Heterogeneity was evaluated by the *I^2^* value and Cochran’s Q. The *I^2^* value was used to assess the heterogeneity of the studies as follows: 0%–39%, low heterogeneity; 40%–74%, moderate heterogeneity; and 75%–100%, high heterogeneity.

All meta-analyses were conducted using open-source statistical software (Review Manager Version 5.3. Copenhagen: The Nordic Cochrane Centre, The Cochrane Collaboration, 2014). All *p* values are two-sided, and *p* < 0.05 was considered statistically significant. Calculations were performed using commercial software (SPSS version 20, IBM Inc; Armonk NY, USA)

## 3. Results

### 3.1. Literature Search and Data Extraction

The search strategy yielded 2543 potentially eligible studies from the PubMed and Cochrane Library databases and 22 studies by hand-search through other sources and papers (Figure 1). Forty studies were selected from the extracted studies. Of those, six studies involved non-PAC and/or non-PAM regimens, three were duplicated studies, two were reviews, and two studies had treatment periods > 14 days; these were excluded. Ultimately, a total of 27 full articles were assessed for eligibility (Figure 1) [17,18,23,27,28,29,30,31,32,33,34,35,36,37,38,39,40,41,42,43,44,45,46,47,48,49,50] and a total of 4825 patients treated with the PAC vs. PAM regimens for *H. pylori* infection were included in the analysis. 

The characteristics of the trials are shown in Table 1. Mean patient age ranged from 26–77 years (Table 1). Eradication therapy was performed by a regimen including PPI (pantoprazole (40 mg, twice-daily dosing (bid), omeprazole (20 mg, bid), lansoprazole (30 mg, bid), rabeprazole (10 or 20 mg, bid) or esomeprazole (20 mg, bid)) and AMPC (500, 750 or 1,000 mg, bid, or 500 mg, three times daily (tid)). In addition, the PAC therapy used CAM (200, 250, 400, or 500 mg, bid) and the PAM therapy used any imidazole antimicrobial agent (MNZ: 250, 400, 500, or 750 mg bid, or 400 mg tid, or tinidazole, 500 mg bid). Most of the studies administered PAC and PAM therapy for seven days, but two studies used 14 days [40,50] and two used 7–14 days (Table 1) [43,45]. 

Although adherence is important to receive high eradication rates, there is no significant different adherence between the PAC therapy and the PAM therapy in each study (Appendix A).

### 3.2. Meta-Analysis for Eradication Rate of PAC vs. PAM Therapy

When all 27 trials were analyzed, the ITT eradication rates of the PAC therapy and PAM therapy groups were 74.8% (1833/2451, 95% confidence interval (CI): 73.0%–76.5%) and 72.5% (1721/2374, 95% CI 70.9%–74.4%) and the PP eradication rates were 81.3% (1823/2242, 95% CI: 79.6%–82.9%) and 78.6% (1705/2168, 95% CI 76.9%–80.4%), respectively (*P* =0.08 and 0.06). Meta-analysis showed that the relative risk (RR) for successful eradication with PAC therapy compared with PAM therapy was 1.13 (95% CI: 0.91–1.39, *p* = 0.27) in the ITT analysis (Figure 2A) and 1.18 (95% CI: 0.90–1.55, *p* = 0.22) in the PP analysis (Figure 2B) in the random-effects model; and 1.07 (95% CI: 0.98–1.17, *p* = 0.14) and 1.11 (95% CI: 0.99–1.24, *p* = 0.08) in the fixed-effects model (Appendix A). There was high heterogeneity among these studies (ITT analysis: *p* < 0.01; *I*^2^ = 75%, the PP analysis: *p* <0.01; *I*^2^ = 76%). The funnel plot of all included studies showed asymmetry between PAC therapy and PAM therapy (Figure 3).

### 3.3. Subgroup Analysis of Eradication Rates by Different Geographical Areas Based on Rates of CAM and MNZ Resistance

We assessed resistance to CAM and MNZ using previous studies [10,11,23,24,25,26] (Appendix A). Twelve studies documented areas with high resistance to both CAM (≥ 15%) and MNZ (≥ 15%) (MNZ-R/CAM-R), namely: China, Korea, Morocco, Poland, Taiwan, Turkey, and the United Kingdom [28,29,31,32,33,37,38,39,41,42,43,44,46,47,49]. Areas with high resistance to MNZ and low resistance to CAM (MNZ-R/CAM-S) included Tunisia, North India, Italy, Germany and Finland, in nine studies [1,5,8,9,10,13,14,16,27]. Japan was classified as an area with low MZN resistance and high CAM resistance (MNZ-S/CAM-R) in four studies [23,28,31,34,35,36,39,40,46]. There was no area with low resistance to both CAM and MNZ. 

On analysis of studies from the MNZ-R/CAM-S area, eradication rates for PAC and PAM therapy were 72.9% (481/660, 95% CI: 69.3%–76.2%) and 67.2% (433/644, 95% CI: 63.5%-70.9%), respectively, in the ITT analysis. Meta-analysis showed that the RR for successful eradication by PAC therapy compared to PAM therapy was 1.26 (95% CI: 0.95–1.67, *P* = 0.11) in the random-effects model (Figure 4A), and 1.47 (95% CI: 1.17–1.85, *p* < 0.01) in the fixed-effects model (Appendix A). There was moderate heterogeneity among these studies (*p* = 0.02; *I*^2^ = 57%).

On analysis of studies from the MNZ-S/CAM-R area, eradication rates were 70.8% (240/339, 95% CI: 65.6%–75.6%) for PAC and 92.5% (347/375, 95% CI: 89.4%–95.0%) for PAM therapy in the ITT analysis. In meta-analysis, the RR for successful eradication with PAC therapy compared to PAM therapy was 0.29 (95% CI: 0.13–0.68, *p* <0.01) in the random-effects model (Figure 4B) and 0.23 (95% CI: 0.15–0.36, *p* <0.01) in the fixed-effects model (Appendix A). There was high heterogeneity among these studies (*p* < 0.01; *I*^2^ = 76%).

In the MNZ-R/CAM-R area, eradication rates were 72.4% (668/923, 95% CI: 69.4%-75.2%) for PAC therapy and 65.3% (610/934, 95% CI 62.2%–68.4%) for PAM therapy in the ITT analysis. The RR of successful eradication of PAC therapy compared to PAM therapy was 1.37 (95% CI: 1.05–1.77, *p* = 0.02) in the random-effects model (Figure 4C) and 1.22 (95% CI: 1.07–1.40, *p* < 0.01) in the fixed-effects model (Appendix A). There was moderate heterogeneity among these studies (*p* < 0.01; *I*^2^ = 66%).

The funnel plots of studies from the MNZ-R/CAM-S area (Figure 5A), the MNZ-S/CAM-R area (Figure 5B) and the MNZ-R/CAM-R area (Figure 5C) suggest asymmetry between PAC therapy and PAM therapy.

When we divided studies into different kinds of PPI, PAM therapy had a significantly higher eradication rate compared with the PAC therapy in patients living in the area of MNZ-S/CAM-R. Eradication rates with PAC therapy were < 75%, and the eradication rate in studies from the MNZ-R/CAM-S area was similar to that with PAM therapy in spite of different kinds of PPI (data not shown). However, because number of studies was small, we could not perform sub-analysis by different treatment duration (seven days and 10/14 days) and different daily dose of drugs (PPI, amoxicillin, clarithromycin and metronidazole). 

### 3.4. Meta-Analysis of the Incidence Rates of Adverse Events from PAC Therapy vs PAM Therapy

Nineteen studies (70.4%) provided information on adverse events, namely diarrhea, skin rash, dysgeusia, and nausea (Table 2) [18,23,27,29,31,32,33,34,35,36,37,38,40,42,45,48,49,50]. The incidence of adverse events from PAC therapy vs. PAM therapy were 22.8% (308/1351, 95% CI: 20.6%–25.1%) and 22.6% (317/1402, 95% CI 20.4%–24.9%), respectively. There was no difference in the incidence of adverse events between the two regimens (RR: 1.03, 95% CI: 0.91–1.17, *P* = 0.61) in the random-effects model (Figure 6). There was no heterogeneity among these studies (*p* = 0.49; *I*^2^ = 0%).

The incidence rate of diarrhea was similar between PAC therapy (8.9%, 144/1623, 95% CI: 7.5%–10.4%) and PAM therapy (6.9%, 109/1578, 95% CI: 5.7%-8.3%) (RR: 0.84, 95% CI: 0.67–1.05, *p* = 0.13). The incidence rates of skin rash and dysgeusia were also similar. In contrast, the incidence rate of nausea with PAM therapy (5.9%, 78/1324, 95% CI: 4.7%–7.3%) was significantly higher than that with PAC therapy (2.7%, 37/1368, 95% CI: 1.9%–3.7%) (RR: 2.07, 95% CI: 1.42–3.02, *p* < 0.01).

## 4. Discussion

This meta-analysis of 27 RCTs reevaluated the efficacy and safety of PAC therapy and PAM therapy. The overall eradication rate for PAC therapy and PAM therapy was 74.8% (95% CI: 73.2%–76.5%) and 72.5% (95% CI: 70.6–74.3), respectively. However, in the sub-analysis, PAM therapy had a significantly higher eradication rate, > 90%, compared with PAC therapy in patients living in the area of MNZ-S/CAM-R. Therefore, the PAM regimen may be selected for patients living in the MNZ-S/CAM-R area to achieve a high eradication rate. On the other hand, overall eradication rates with PAC therapy were insufficient, < 75%, and the eradication rate in studies from the MNZ-R/CAM-S area was surprisingly similar to that with PAM therapy. First-line therapy for *H. pylori* eradication should have an eradication rate > 90% [51]; accordingly, the low rate for PAC therapy (< 90%) may be insufficient, irrespective of susceptibility to CAM.

### 4.1. Importance of H. pylori Antimicrobial Agent Resistance

In general, diagnosis and treatment for infectious diseases require culture and sensitivity testing for antimicrobial agents. *H. pylori* is a gram-negative bacterium; the international treatment guidelines for *H. pylori* infection recommend that susceptibility to antimicrobial agents be checked using several tests [51,52]. Culture test and real-time polymerase chain reaction (PCR) methods are generally used to evaluate susceptibility to antimicrobial agents using clinical samples of gastric mucosa, gastric juice, and stool samples [53,54]. Recently, the efficacy of tailored treatment based on sensitivity to CAM (PAC therapy for patients infected with CAM-sensitive strains and PAM therapy for patients infected with CAM-resistant strains) has been shown, with eradication rates > 90% [55,56]. Although this tailored treatment based on susceptibility to CAM had a high eradication rate, especially in areas with a high incidence rate of CAM-resistant *H. pylori*, tailored treatment for all patients with *H. pylori* infection or all areas/populations in the world is not feasible, due to disadvantages in culture testing and PCR in clinical practice, as we describe below. Collection of biopsy specimens and culture of gastric mucosa via endoscopy increase the risk of hemorrhage, especially in patients taking antithrombotic drugs [57]. Culture is also inconvenient and time consuming. In addition, importantly, when comparing with standard triple therapy, the cost-effectiveness of tailored treatment guided by culture and PCR should be understood. However, because our results showed that the efficacy of PAC and PAM therapies varied among countries depending on the prevalence of CAM- and MNZ-resistant strains, it is also important to clarify changes in the incidence of drug resistance among locations [10].

In this meta-analysis, PAM therapy showed high efficacy in populations in areas with a low incidence of MNZ resistance (i.e., Japan), and would accordingly be recommended as first-line therapy in these populations. Because few areas have low (< 15%) MNZ resistance, Japan may be distinctive from the perspective of susceptibility to MNZ. Although MNZ is widely used worldwide as a treatment for anaerobic bacteria in both children and adults, the Japanese health insurance system initially approved MNZ for adult trichomoniasis only. Therefore, although the Japanese health insurance system approved second-line PAM therapy for patients with *H. pylori* infection, our findings suggest that PAM be used as first-line therapy. Although a similar susceptibility pattern was not seen, PAM therapy may be effective in regions with low MNZ resistance (< 20%; i.e., North America [58] or Chile [59]).

Meanwhile, eradication rates with PAC therapy were extremely low not only in areas with CAM/MNZ-double resistant *H. pylori* and but also in CAM-sensitive areas, at only 73% (ITT analysis) and 82% (PP analysis). Although PAC therapy is recommended for regions with CAM resistance rates < 15%, our present results show that PAC therapy is insufficient as a first-line therapy even in regions with low resistance. What is the reason for this low eradication rate when CAM resistance rate is generally low (< 15%)? We previously showed high efficacy (> 90%) in tailored treatment based on susceptibility to CAM [56], and this was also shown in a study from Finland with low CAM resistance (2%), with an eradication rate > 90% with PAC therapy (Appendix A) [23]. Thus, we recommend that the country incidence of CAM resistance be re-evaluated when PAC therapy is used, or a decrease in the cut-off value of CAM resistance rate to < 5–10%. In addition, since the incidence of CAM resistance to *H. pylori* is expected to increase in the future, alternative optimal treatment regimens will be required for patients with *H. pylori* infection living in CAM-R/MNZ-R or CAM-S/MNZ-R areas. 

### 4.2. Alternatives to Triple Therapy

Bismuth and non-bismuth quadruple therapies have been recommended as alternative eradication strategies [1]. Bismuth quadruple therapy, consisting of bismuth, tetracycline, metronidazole, and amoxicillin, is traditionally used and is an acceptable therapy worldwide [60,61]. However, meta-analyses comparing it with triple conventional therapy showed that bismuth quadruple therapy offered no advantage over triple therapy in either eradication rate or prevention of adverse events [62,63,64]. In addition, a disadvantage of bismuth quadruple therapy is its more complicated dosing schedule, which makes adherence difficult to maintain. Some countries, such as Japan, France, and Australia, cannot use bismuth preparations due to a lack of approval or availability [65], and the possibility of it becoming a universal treatment may accordingly be low. However, because bismuth quadruple therapy revealed high eradication efficacy in regions with a high incidence of both MNZ and CAM resistance [66], this regimen may be indicated in limited populations (i.e., patients infected with MNZ- and CAM-resistant strains).

Non-bismuth quadruple therapy (concomitant therapy), which consists of a PPI combined with amoxicillin, clarithromycin, and a nitroimidazole bid for 5–10 days, has achieved an eradication rate of 91.7% in patients infected with CAM-sensitive strains [67]. In areas with CAM-S/MNZ-R, this regimen may be optimal [68], as the eradication rate with concomitant therapy in patients with double CAM- and MNZ-resistant strains was only 33.3%–66.7% [66]. Clinicians will need to watch for adverse events, such as nausea, headache and dizziness, which are more frequent than with triple therapy due to the intake of three antimicrobial agents at the same time [67,69]. 

### 4.3. Efficacy of Vonoprazan-Containing Triple Therapy

High intragastric pH levels during eradication therapy improved eradication rates for *H. pylori* infection [70,71,72]. Vonoprazan (VPZ), a potent new acid secretion inhibitor, inhibits gastric parietal cell potassium ions binding to H^+^/K^+^ -ATPase more potently and more reliably than conventional PPIs, suggesting it may be more effective [73,74,75]. A number of RCTs have compared eradication rates of VPZ-containing triple therapy and conventional PPI-containing triple therapy [76,77,78]. In 2016, a phase III study in 650 *H. pylori*-positive subjects confirmed that the first-line eradication rate for triple therapy containing VPZ, AMPC, and CAM was 92.6%, and thus higher than that for PAC therapy including lansoprazole (75.9%) [76]. A previous meta-analysis also showed that VPZ-containing triple therapy had a significantly higher eradication rate than PAC therapy (88.1% and 72.8%) [79], especially in patients infected with CAM-resistant strains (82.0% vs. 40.0%) [80]. However, in patients infected with CAM-sensitive strains, a meta-analysis showed no significant difference in eradication rates between VPZ-containing triple therapy and PAC therapy (95.3% and 92.8%), suggesting that the efficacy of VPZ-containing triple therapy was limited for patients infected with CAM-resistant strains. Therefore, VPZ-containing triple therapy has potential in populations or regions with well-documented high CAM resistance (more than 15%), such as the MNZ-S/CAM-R and MNZ-R/CAM-R areas. The efficacy of VPZ-containing triple therapy has not yet been compared to that of PAM, bismuth quadruple, and non-bismuth quadruple therapies in regions with high CAM resistance. It has also not yet been shown whether VPZ-based triple therapy containing MNZ has greater efficacy than PAM therapy. 

## 5. Conclusions

In conclusion, eradication rates and adverse event rates of PAC and PAM therapies were found to be equivalent worldwide. However, eradication rates of both regimens were insufficient to eradicate *H. pylori* infection without considering drug resistance in more than 90% of *H. pylori*-positive patients. The results of this meta-analysis suggest that because the efficacy of eradication therapy is based on susceptibility to antimicrobial agents, PAM therapy may have the potential to eradicate > 90% of *H. pylori* infections in patients infected with MNZ-sensitive strains, whereas the eradication rate of PAC therapy is low even in patients infected with CAM-sensitive strains. The incidence of antimicrobial agent resistance in *H. pylori* is changing among geographic areas, and it is therefore important to periodically reevaluate the efficacy of eradication regimens.

## Figures and Tables

**Figure 1 jcm-09-00543-f001:**
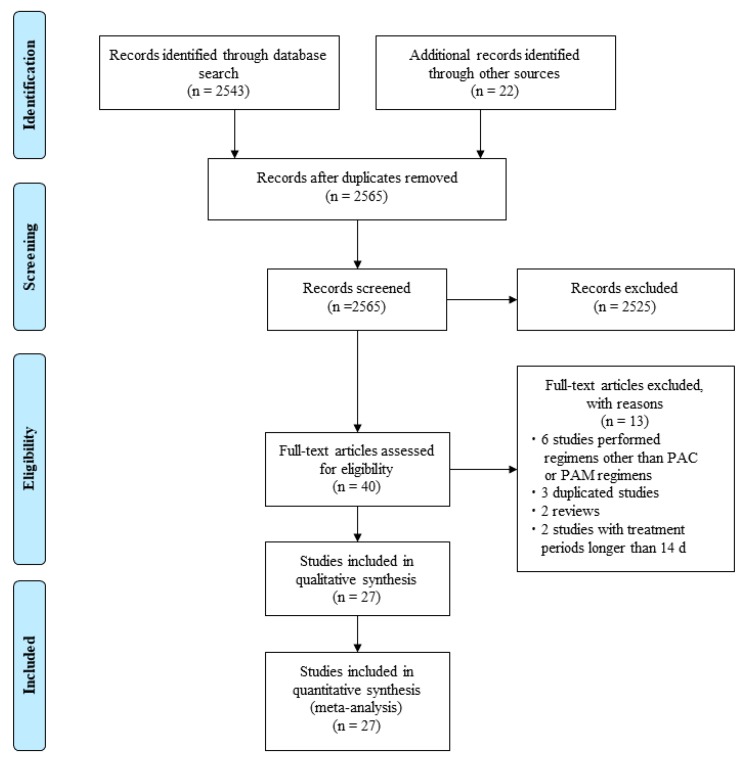
Workflow for the selection of studies comparing eradication rates between PAC and PAM therapy. Abbreviations: D, day; PAC, proton pump inhibitor/amoxicillin/clarithromycin; PAM, proton pump inhibitor/amoxicillin/metronidazole.

**Figure 2 jcm-09-00543-f002:**
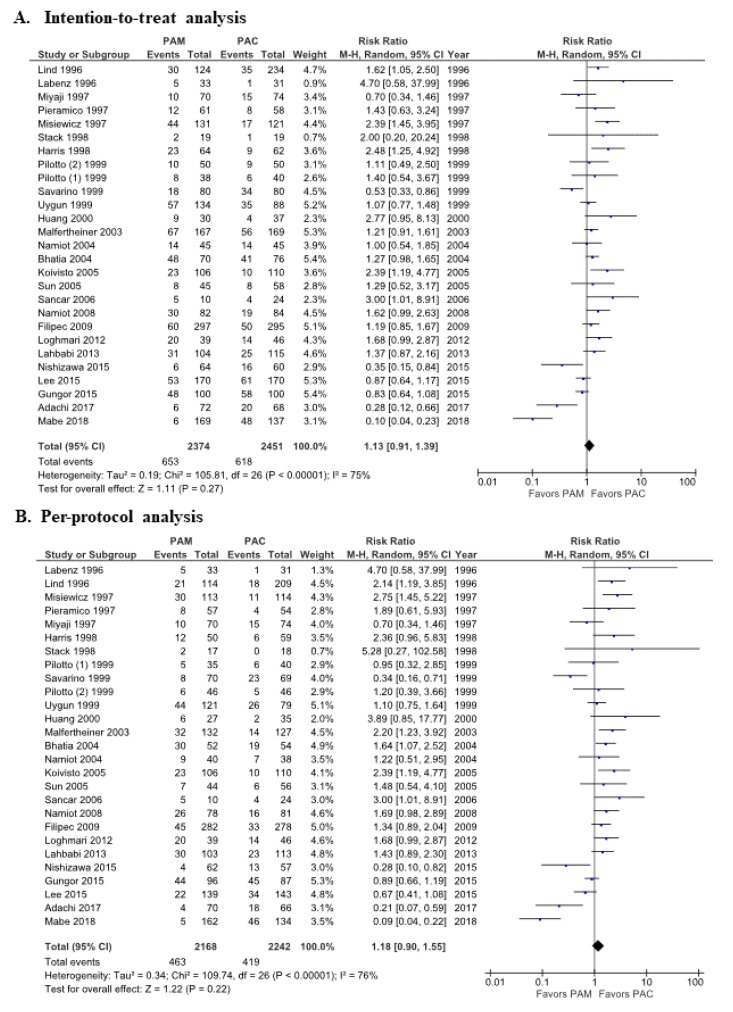
Forest plots of overall eradication rates between PAM therapy and PAC therapy in all eligible trials with intention-to-treat analysis (**A**) and per-protocol analysis (**B**) in the random-effects model. The relative risk (RR) for successful eradication between PAC therapy and PAM therapy was similar in the intention-to-treat (ITT) analysis (**A**) and the per-protocol (PP) analysis (**B**). Abbreviations: CI, confidence interval; PAC, proton pump inhibitor (PPI)/amoxicillin/clarithromycin; PAM, PPI/amoxicillin/metronidazole.

**Figure 3 jcm-09-00543-f003:**
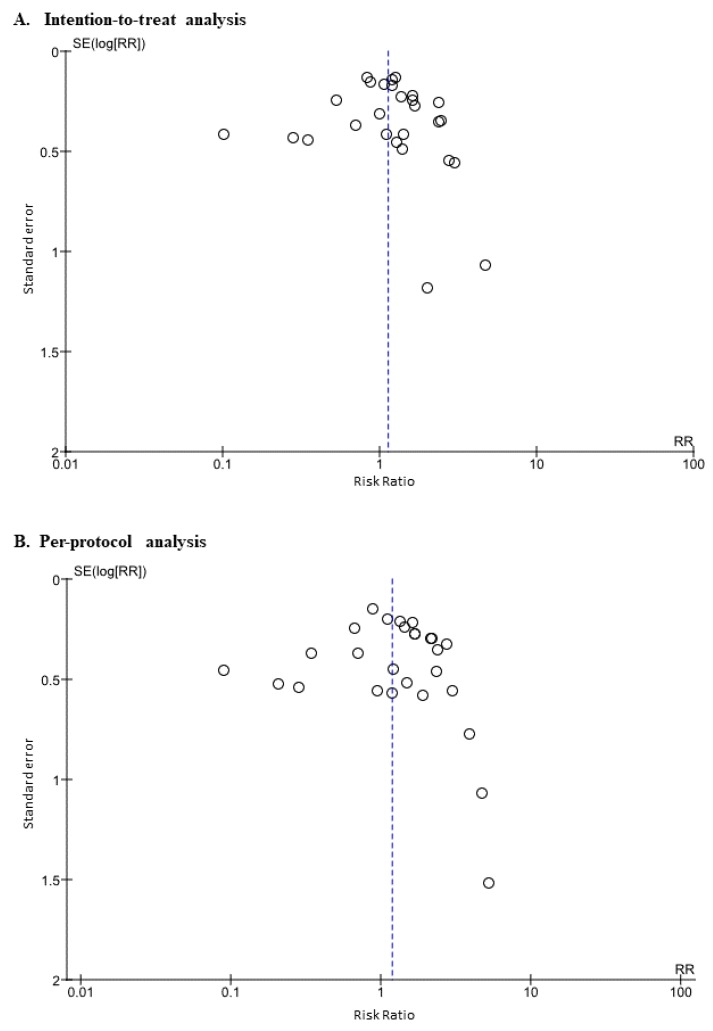
Funnel plot for the comparison of efficacy between PAC therapy and PAM therapy on intention-to-treat analysis (**A**) and per-protocol analysis (**B**). Abbreviation: PAC, proton pump inhibitor (PPI)/amoxicillin/clarithromycin; PAM, PPI/amoxicillin/metronidazole; RR, risk ratio; SE, standard error.

**Figure 4 jcm-09-00543-f004:**
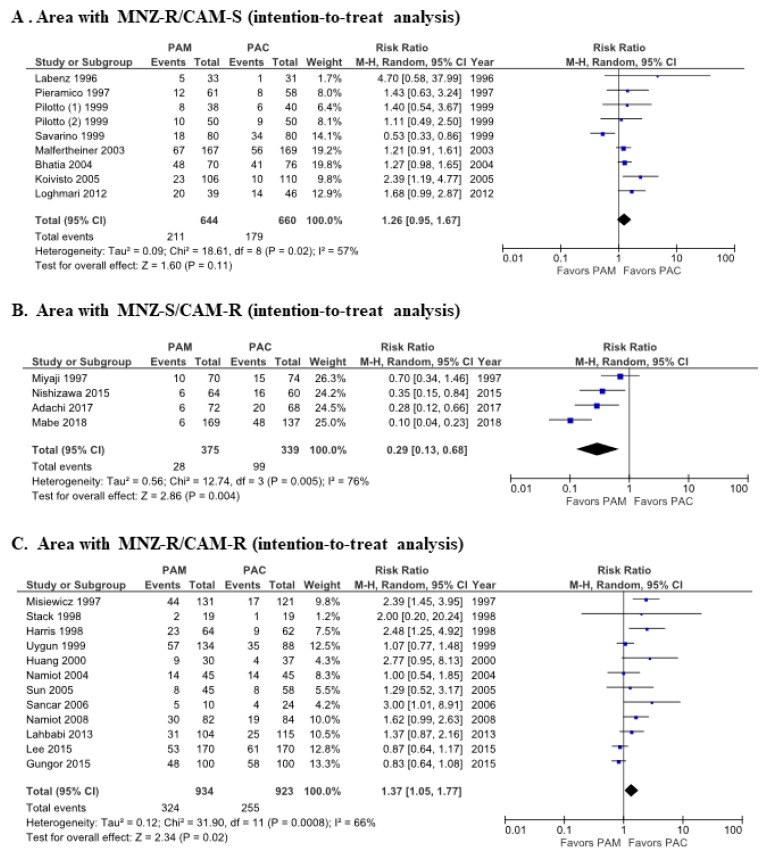
Forest plots of eradication rates between PAM therapy and PAC therapy in areas classified as having (**A**) low clarithromycin resistance (< 15%) and high metronidazole (≥ 15%) resistance, (**B**) high clarithromycin resistance and low metronidazole resistance, and (**C**) high clarithromycin resistance and high metronidazole resistance in intention-to-treat analysis. Abbreviations: CAM, clarithromycin; CI, confidence interval; MNZ, metronidazole; PAC, proton pup inhibitor (PPI)/amoxicillin/clarithromycin; PAM, PPI/amoxicillin/metronidazole.

**Figure 5 jcm-09-00543-f005:**
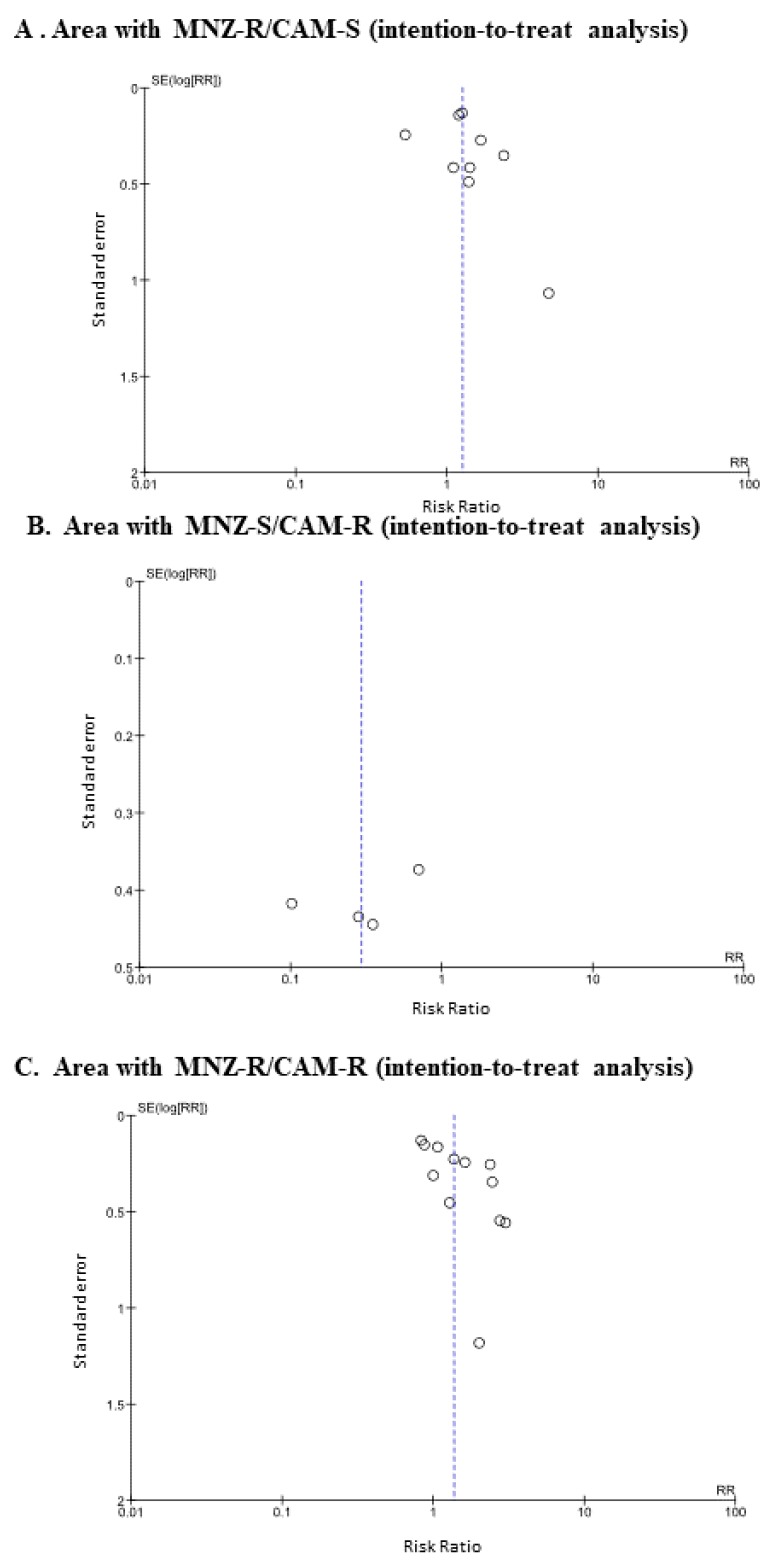
Funnel plots of efficacy between PAM therapy and PAC therapy in areas classified as having (**A**) low CAM resistance (< 15%) and high MNZ resistance (≥ 15%), (**B**) high CAM resistance and low MNZ resistance, and (**C**) high CAM resistance and high MNZ resistance in intention-to-treat analysis. Abbreviations: CAM, clarithromycin; MNZ, metronidazole; PAC, proton pup inhibitor (PPI)/amoxicillin/clarithromycin; PAM, PPI/amoxicillin/metronidazole; RR, risk ratio; SE, standard error.

**Figure 6 jcm-09-00543-f006:**
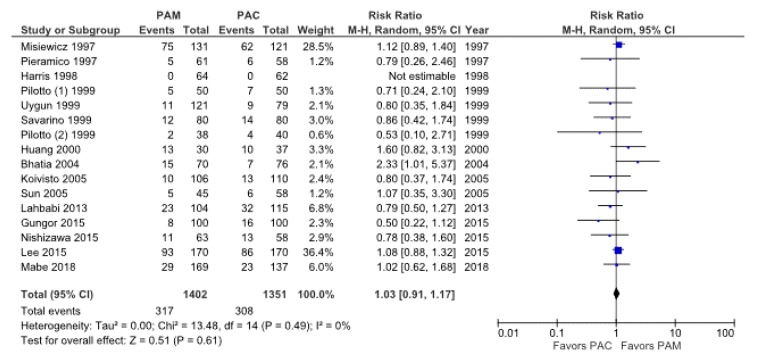
Forest plot of the incidence rate of all adverse events between PAC therapy and PAM therapy in the random effects model. Abbreviation: CI, confidence interval; PAC, proton pump inhibitor (PPI)/amoxicillin/clarithromycin; PAM, PPI/amoxicillin/metronidazole.

**Table 1 jcm-09-00543-t001:** Characteristics of the studies comparing efficacy of PAC vs. PAM regimens.

Authors	Year	Country	Patient Number	Age (mean)	Sex (M/F)	Regimen of PAC Therapy (Dosing Dose a Day)	Duration	Eradication Rate of PAC (ITT/PP)	Regimen of PAM Therapy (Dosing Dose a Day)	Duration	Eradication Rate of PAM (ITT/PP)
Labenz et al. [28]	1996	Germany	64	NA	NA	OPZ (20), AMPC (1000), CAM (500)	7 days	96.8%/96.8%	OPZ (20), AMPC (1000), MNZ (500)	7 days	84.8%/84.8%
Lind et al. [27]	1996	Multiple	391	NA	NA	OPZ (20), AMPC (1000), CAM (500)	7 days	85.0%/91.4%	OPZ (20), AMPC (1000), MNZ (400)	7 days	75.8%/81.6%
Misiewicz et al. [29]	1997	UK	252	48	176/76	LPZ (30), AMPC (1000), CAM (250)	7 days	86.0%/90.4%	LPZ (30), AMPC (1000), MNZ (400)*	7 days	66.4%/73.5%
Miyaji et al. [30]	1997	Japan	144	52	NA	PPIs AMPC (500), CAM (200)	7 days	79.7%/79.7%	PPIs AMPC (500), MNZ (250)	7 days	85.7%/85.7%
Pieramico et al. [31]	1997	Italy	119	53	61/58	OPZ (20), AMPC (1000), CAM (500)	7 days	86.2%/92.6%	OPZ (20), AMPC (1000), MNZ (500)	7 days	80.3%/86.0%
Stack et al. [32]	1997	UK	38	52	28/10	RPZ (20), AMPC (1000), CAM (500)	7 days	94.7%/100.0%	RPZ (20), AMPC (1000), MNZ (400)	7 days	89.5%/88.2%
Harris et al. [33]	1998	UK	126	47	88/38	PPIs AMPC (1000), CAM (250)	7 days	85.5%/89.8%	PPIs AMPC (500), MNZ (400)	7 days	64.1%/76.0%
Pilotto et al. [34]	1999	Italy	100	77	43/57	LPZ (30), AMPC (1000), CAM (250)	7 days	82.0%/89.1%	LPZ (30), AMPC (1000), MNZ (250)	7 days	80.0%/87.0%
Pilotto et al. [35]	1999	Italy	78	71	28/50	PAN (40), AMPC (1000), CAM (250)	7 days	85.0%/85.0%	PAN (40), AMPC (1000), MNZ (250)	7 days	78.9%/85.7%
Savarino et al. [36]	1999	Italy	160	49	86/74	OPZ (20), AMPC (1000), CAM (250)	7 days	57.5%/66.7%	OPZ (20), AMPC (1000), MNZ (500)	7 days	77.5%/88.6%
Uygun et al. [37]	1999	Turkey	222	39	119/103	OPZ (20), AMPC (1000), CAM (500)	7 days	60.2%/67.1%	OPZ (20), AMPC (1000), MNZ (500)/TNZ (500)	7 days	57.5%/63.6%
Huang et al. [38]	2000	Taiwan	67	42	37/30	LPZ (30), AMPC (1000), CAM (500)	7 days	89.2%/94.3%	LPZ (30), AMPC (1000), MNZ (500)	7 days	70.0%/77.8%
Malfertheiner et al. [39]	2003	Germany	336	57	184/152	PAN (40), AMPC (1000), CAM (500)	7 days	66.9%/89.0%	PA (40), AMPC (1000), MNZ (500)	7 days	59.9%/75.8%
Bhatia et al. [40]	2004	India	146	39	113/33	LPZ (30), AMPC (1000), CAM (500)	14 days	46.1%/64.8%	LPZ (30), AMPC (1000), TNZ (500)	14 days	31.4%/42.3%
Namiot et al. [41]	2004	Poland	90	42	68/22	OPZ (20), AMPC (1000), CAM (250)	7 days	68.9%/81.6%	OPZ( 20), AMPC (1000), TNZ (500)	7 days	68.9%/77.5%
Koivisto et al. [23]	2005	Finland	216	56	103/113	LPZ (30), AMPC (1000), CAM (500)	7 days	90.9%/90.9%	LPZ (30), AMPC (1000), MNZ (400)*	7 days	78.3%/78.3%
Sun et al. [42]	2005	China	103	51	85/18	OPZ (20), AMPC (1000), CAM (500)	7 days	86.2%/89.3%	OPZ (20), AMPC (1000), MNZ (500)	7 days	82.2%/84.1%
Sancar et al. [43]	2006	Turkey	35	43	18/17	LPZ (30)/OPZ (20), AMPC (1000), CAM (500)	7, 14 days	83.3%/83.3%	LPZ (30)/OPZ (20), AMPC (1000), MNZ (500)	7, 14 days	50.0%/50.0%
Namiot et al. [44]	2008	Poland	159	50	106/53	OPZ( 20), AMPC (1000), CAM (250)	7 days	77.4%/80.2%	OPZ (20), AMPC (1000), MNZ (250)	7 days	63.4%/66.7%
Filipec et al. [45]	2009	Croatia	592	52	335/257	PAN (40), AMPC (1000), CAM (500)	7, 10, 14 days	83.1%/88.1%	PAN (40), AMPC (1000), MNZ (500)	7, 10, 14 days	79.8%/84.0%
Loghmari et al. [46]	2012	Tunisia	85	40	44/41	OPZ (20), AMPC (1000), CAM (500)	7 days	69.6%/69.6%	OPZ (20), AMPC (1000), MNZ (500)	7 days	48.7%/48.7%
Lahbabi et al. [47]	2013	Morocco	219	47	105/114	PPIs AMPC (1000), CAM (500)	7 days	78.3%/79.6%	PPIs AMPC (1000), MNZ (500)	7 days	70.2%/70.9%
Nishizawa et al. [48]	2015	Japan	219	47	105/114	PAN (40), AMPC (1000), CAM (500)	14 days	42.0%/48.3%	PAN (40), AMPC (1000), MNZ (500)	14 days	52.0%/54.2%
Lee et al. [49]	2015	Korea	340	57	209/131	RPZ (20), AMPC (1000), CAM (500)	7 days	64.1%/76.2%	RPZ (20), AMPC (1000), MNZ (750)	7 days	68.8%/84.2%
Gungor et al. [50]	2015	Turkey	124	61	51/73	RPZ (10), AMPC (750), CAM (400)	7 days	73.3%/77.2%	RPZ (10), AMPC (750), MNZ (250)	7 days	90.6%/93.5%
Adachi et al. [17]	2017	Japan	140	64	68/72	EPZ (20), AMPC (750), CAM (400)	7 days	70.6%/72.7%	EPZ (20), AMPC (750), MNZ (500)	7 days	91.7%/94.3%
Mabe et al. [18]	2018	Japan	306	26	157/149	LPZ (30), AMPC (750), CAM (200)	7 days	65.0%/65.7%	LPZ (30), AMPC (750), MNZ (250)	7 days	96.4%/96.9%

Abbreviations: AMPC, amoxicillin; CAM, clarithromycin; D, day: EPZ, esomeprazole; LPZ, lansoprazole; MNZ, metronidazole; NA, not available; OPZ, omeprazole; PAC, proton pump inhibitor/amoxicillin/clarithromycin; PAM, proton pump inhibitor/amoxicillin/metronidazole; PAN, pantoprazole; PPI, proton pump inhibitor; RPZ, rabeprazole; TNZ, Tinidazole; UK, United Kingdom.

**Table 2 jcm-09-00543-t002:** Adverse effects of PAC and PAM therapy in individual studies.

Authors	PAC						PAM					
	*N*	Events	Diarrhea	Skin Rash	Dysgeusia	Nausea	*N*	Events	Diarrhea	Skin Rash	Dysgeusia	Nausea
Lind et al. [27]	234	NA	29	NA	18	2	124	NA	24	NA	11	2
Misiewicz et al. [29]	121	51.2%	22	NA	2	NA	131	57.3%	19	NA	13	NA
Pieramico et al. [31]	58	10.3%	2	0	1	0	61	8.2%	1	0	0	1
Stack et al. [32]	19	NA	9	NA	8	NA	19	NA	11	NA	3	NA
Harris et al. [33]	62	0.0%	0	0	0	0	64	0.0%	0	0	0	0
Pilotto et al. [34]	50	14.0%	NA	NA	NA	NA	50	10.0%	NA	NA	NA	NA
Pilotto et al. [35]	40	10.0%	NA	NA	NA	NA	38	5.3%	NA	NA	NA	NA
Savarino et al. [36]	80	17.5%	3	1	4	3	80	15.0%	1	2	2	3
Uygun et al. [37]	79	11.4%	3	0	5	0	121	9.1%	1	2	0	7
Huang et al. [38]	37	27.0%	3	0	1	3	30	43.3%	2	0	4	3
Bhatia et al. [40]	76	9.2%	NA	NA	NA	NA	70	21.4%	NA	NA	NA	NA
Koivisto et al. [23]	110	11.8%	NA	NA	NA	NA	106	9.4%	NA	NA	NA	NA
Sun et al. [42]	58	10.3%	1	1	0	1	45	11.1%	1	1	2	1
Filipec et al. [45]	295	NA	13	NA	15	12	297	NA	11	NA	22	26
Lahbabi et al. [47]	115	27.8%	11	1	11	NA	104	22.1%	9	1	6	NA
Nishizawa et al. [48]	100	16.0%	11	1	7	0	100	8.0%	9	1	6	0
Lee et al. [49]	170	50.6%	24	0	28	12	170	54.7%	2	0	10	32
Gungor et al. [50]	58	22.4%	5	5	1	2	63	17.5%	6	3	0	0
Mabe et al. [18]	137	16.8%	8	4	4	2	169	17.2%	12	4	4	3

Abbreviations: N, number; NA, not available; PAC, proton pup inhibitor (PPI)/amoxicillin/ clarithromycin; PAM, PPI/amoxicillin/metronidazole.

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
