# Peer review of "Clarithromycin Versus Metronidazole in First-Line Helicobacter Pylori Triple Eradication Therapy Based on Resistance to Antimicrobial Agents: Meta-Analysis"

_jcm, 2020, doi:10.3390/jcm9020543_

Round 1

Reviewer 1 Report

In this paper, the autors performed a meta-analysis to reevaluate the efficacy and safety of PAC and PAM therapies in relation to CAM/MNZ resistance-defined location. They conclude that eradication rates with PAC and PAM were equivalent worldwide. In low-MNZ-resistance areas, PAM may be recommended as first-line therapy. However, the efficacy of PAC may be insufficient, irrespective of susceptibility to CAM.

Overall the manuscript is well organized and well written.

Author Response

Our responses to comments raised by the Reviewer 1

In this paper, the autors performed a meta-analysis to reevaluate the efficacy and safety of PAC and PAM therapies in relation to CAM/MNZ resistance-defined location. They conclude that eradication rates with PAC and PAM were equivalent worldwide. In low-MNZ-resistance areas, PAM may be recommended as first-line therapy. However, the efficacy of PAC may be insufficient, irrespective of susceptibility to CAM.

Overall the manuscript is well organized and well written.

Response:

Thank you for you comments. This study investigated efficacy and safety of PAC and PAM therapies in relation with clarithromycin/metronidazole resistance-defined location. We believe that this meta-analysis will be of a great interest to the readers of J Clin Med.

Reviewer 2 Report

This is a well-written meta-analysis comparing clarithromycin versus metronidazole in first-line H. pylori triple therapy. The authors point out well the rationale for performing the study as well as present their methods and results in a concise and understandable way. However, some of the conclusions drawn are over-interpreted in my opinion and further subgroup-analyses as pointed out below should be considered before suggesting area-specific treatment regimens.

1) From table 1 it appears that eradication rates may correlate with drug regimen. The authors may consider analyzing whether the effect they see by area is independent of regimen (and duration of treatment).

2) Another point important for successful eradication is adherence of the patients. I understand that this is most likely not assessed in the included studies but should be at least discussed in this manuscript by the authors (or if data available considered for additional analysis).

Consequently, the statement in lines 241, 242 should be phrased more cautious as for example given in the conclusion of the abstract.

Similarly, the conclusion in line 306 should be revised, there is no evidence in this manuscript given to provide such a recommendation.

Author Response

Our responses to comments raised by the Reviewer 2

From Table 1 it appears that eradication rates may correlate with drug regimen. The authors may consider analyzing whether the effect they see by area is independent of regimen (and duration of treatment).

Response

Thank your for your suggestions. According to your comments, we re-analyzed effects based on incidence rates of CAM-R/MNZ-R H. pylori strains as sub-analysis.

Our finding that “PAM therapy had a significantly higher eradication rate of more than 90% compared with PAC therapy in patients living in the area of MNZ-S/CAM-R” and “Overall eradication rates with PAC therapy were insufficient, < 75%, and the eradication rate in studies from the MNZ-R/CAM-S area was similar to that with PAM therapy” was shown in spite of different kinds of PPI. However, because number of studies was small, we could not perform sub-analysis by different duration (7 days and 10/14 days) and different daily dose of drugs (PPI, amoxicillin, clarithromycin and metronidazole).

According to your suggestion, we add above results in the “Results” section of revised version.

Another point important for successful eradication is adherence of the patients. I understand that this is most likely not assessed in the included studies but should be at least discussed in this manuscript by the authors (or if data available considered for additional analysis).

Consequently, the statement in lines 241, 242 should be phrased more cautious as for example given in the conclusion of the abstract.

Similarly, the conclusion in line 306 should be revised, there is no evidence in this manuscript given to provide such a recommendation.

Response

We agree with your comments. We also believe that adherence is important to receive high eradication rates. According to your suggestion, we add new table as Table S2 to show adherence in each studies. However, there was no significant difference of adherence between the PAC therapy and the PAM therapy in each study. We add Table (S2) and any comments in the “Results” section of the revised version.

According to your suggestion, the statement in lines 241, 242 was revised in “accordingly, the low rate for PAC therapy (< 90%) may be insufficient, irrespective of susceptibility to CAM”.

In addition, as your comment, in the conclusion in line 306, we have no evidence in this manuscript given to provide such a recommendation. Therefore, we delete this sentence in the revised version.

Round 2

Reviewer 2 Report

I have no further comments.